# Token Merging: Your ViT But Faster

**Daniel Bolya**[1,2*]    **Cheng-Yang Fu**[2]    **Xiaoliang Dai**[2]    **Peizhao Zhang**[2]
**Christoph Feichtenhofer**[2]    **Judy Hoffman**[1]

[1] Georgia Tech    [2] Meta AI

{dbolya,judy}@gatech.edu, {chengyangfu,xiaoliangdai,stzpz,feichtenhofer}@meta.com

## Abstract

We introduce Token Merging (ToMe), a simple method to increase the throughput of existing ViT models *without needing to train*. ToMe gradually combines similar tokens in a transformer using a general and light-weight matching algorithm that is as fast as pruning while being more accurate. Off-the-shelf, ToMe can $2\times$ the throughput of state-of-the-art ViT-L @ 512 and ViT-H @ 518 models on images and $2.2\times$ the throughput of ViT-L on video with only a 0.2-0.3% accuracy drop in each case. ToMe can also easily be applied during training, improving in practice training speed up to $2\times$ for MAE fine-tuning on video. Training with ToMe further minimizes accuracy drop, leading to $2\times$ the throughput of ViT-B on audio for only a 0.4% mAP drop. Qualitatively, we find that ToMe merges object parts into one token, even over multiple frames of video. Overall, ToMe's accuracy and speed are competitive with state-of-the-art on images, video, and audio.

## 1 Introduction

The introduction of transformers (Vaswani et al., 2017) from NLP to vision with Vision Transformers (ViTs) by Dosovitskiy et al. (2020) has rapidly advanced the field of computer vision. However, unlike in NLP, vision has been since dominated by domain-specific transformer hybrids like Swin (Liu et al., 2021; Dong et al., 2022) using vision-specific attention, MViT (Fan et al., 2021; Li et al., 2022) using vision-specific pooling, or LeViT (Graham et al., 2021) using vision-specific conv modules. The reason for this trend is simple: efficiency. Adding vision-specific inductive biases enables transformer hybrids to perform better with less compute.

Yet, vanilla ViTs still have many desirable qualities: they consist of simple matrix multiplications, making them faster than their raw flop count would suggest; they support powerful self-supervised pre-training techniques such as MAE (He et al., 2022) that can put up state-of-the art results while being fast to train; given their lack of assumptions about the data, they can be applied with little or no changes across many modalities (Feichtenhofer et al., 2022; Huang et al., 2022); and they scale well with massive amounts of data (Zhai et al., 2021; Singh et al., 2022), recently obtaining up to 90.94% top-1 on ImageNet (Wortsman et al., 2022).

However, running these massive models can be troublesome, and reproducing these results with a faster architecture would be difficult. A promising subfield of ViTs have recently emerged where, due to the input-agnostic nature of transformers, tokens can be pruned at runtime to enable a faster model (Rao et al., 2021; Yin et al., 2022; Meng et al., 2022; Liang et al., 2022; Kong et al., 2022). Yet, token pruning has several disadvantages: the information loss from pruning limits how many tokens you can reasonably reduce; current methods require re-training the model to be effective (some with extra parameters); most cannot be applied to speed up training; and several prune different numbers of tokens depending on the input content, making batched inference infeasible.

In this work, we present Token Merging (ToMe) to *combine* tokens, rather than prune them. Because of our custom matching algorithm, our method is as fast as pruning while being more accurate. Moreover, our method works *with or without training*, which unlocks its use on huge models with minimal accuracy drop. Using ToMe during training, we observe actual increases in training speed, in some cases cutting the total training time *in half*. And we apply ToMe without any modifications to images, video, and audio and find it to be competitive with the SotA in all cases.

Our contributions are as follows: we introduce a technique to increase the throughput and real-world training speed of ViT models, both with and without training (Sec. 3) and thoroughly ablate our

---

*Work done during an internship at Meta AI. Code at http://github.com/facebookresearch/ToMe

design choices (Sec. 4.1); we perform extensive experiments on images with several ViT models (Sec. 4.2) and compare to state-of-the-art in architecture design and token pruning methods (Sec. 4.3); we then repeat these experiments for both video (Sec. 5) and audio (Sec. 6) and find ToMe works well across modalities; and we visualize our results and find ToMe merges parts of objects on images (Fig. 4) and objects over their entire range of motion on video (Fig. 6). We hope ToMe can enable the creation of more powerful, faster ViT models.

## 2 RELATED WORK

**Efficient Transformers.** Several works have attempted to create more efficient transformers in both NLP and Vision. Some focus on faster attention (Choromanski et al., 2020; Shen et al., 2021; Dao et al., 2022; Wang et al., 2020; Bolya et al., 2022), some attempt to prune heads or features (Meng et al., 2022; Voita et al., 2019; Michel et al., 2019), and some attempt to infuse domain-specific modules (Mehta & Rastegari, 2021; Graham et al., 2021; Liu et al., 2021; 2022a; Dong et al., 2022). In this paper, we focus on speeding up existing ViT models by merging tokens to match the speed-accuracy trade-off of more complicated domain-specific models, sometimes *without training*.

**Token Reduction.** Since transformers can operate with any number of tokens, several recent works have attempted to prune the tokens from transformers in both NLP (Goyal et al., 2020; Kim & Cho, 2020; Kim et al., 2021; Lassance et al., 2021) and Vision (Meng et al., 2022; Yin et al., 2022; Kong et al., 2022; Song et al., 2022; Rao et al., 2021; Fayyaz et al., 2022; Yu & Wu, 2021). However, these methods require training, while our method can be used *without training*. Moreover, most pruning works are *dynamic*, i.e., the number of tokens varies between images or sentences. While this benefits accuracy it limits practicality, as samples with different numbers of tokens can no longer be batched. To solve this, most pruning papers apply a mask during training rather than remove tokens, which negates the speed-up from pruning. Our method, on the other hand, can be applied during both inference and training, achieving real-world speed-ups in either case.

**Combining Tokens.** While plenty of works prune tokens, very few combine them. Kong et al. (2022) and Liang et al. (2022) combine what they prune into a single token. GroupViT (Xu et al., 2022), while not focused on efficiency, groups tokens using cross-attention for semantic segmentation. TokenLearner (Ryoo et al., 2021) uses an MLP to reduce the number of tokens. LIT (Pan et al., 2022) learns deformable token merging layers for pooling between stages. Token Pooling (Marin et al., 2021) is the most similar to our token merging but uses a slow kmeans-based approach[1] that doesn't work on an off-the-shelf model[2]. Until now, no approach has been successful in offering a reasonable speed-accuracy trade-off when combining tokens without training.

## 3 TOKEN MERGING

Our goal is to insert a token merging module into an existing ViT (Dosovitskiy et al., 2020). By merging *redundant* tokens, we hope to increase throughput, while not necessarily having to train.

**Strategy.** In each block of a transformer, we merge tokens to *reduce* by $r$ per layer. Note that $r$ is a quantity of tokens, not a ratio. Over the $L$ blocks in the network, we gradually merge $rL$ tokens. Varying $r$ gives a speed-accuracy trade-off, as fewer tokens means lower accuracy but higher throughput. Importantly, we reduce $rL$ tokens regardless of the image's content. Some pruning methods *dynamically* vary the number of tokens (e.g., Kong et al. (2022)). This increases accuracy but is generally impractical, as it prevents batched inference or training without padding tokens.

As shown in Fig. 1, we apply our token merging step *between* the attention and MLP branches of each transformer block. This is also in contrast to prior works, which tend to place their reduction method at the beginning of the block instead. Our placement allows information to be propagated from tokens that would be merged and enables us to use features within attention to decide what to merge, both of which increase accuracy (see Tab. 1a).

**Token Similarity.** Before merging similar tokens, we must first define what "similar" means. While it may be tempting to call two tokens similar if the distance between their features is small (as in Marin et al. (2021)), this is not necessarily optimal. The intermediate feature space in modern transformers is *overparameterized*. For instance, ViT-B/16 has enough features to completely encode the rgb pixel

---

[1]Their throughput is only 1.14-1.25× the baseline because their method can't be parallelized.
[2]In their appendix, they show drops of 10-40% accuracy when combining tokens without training.

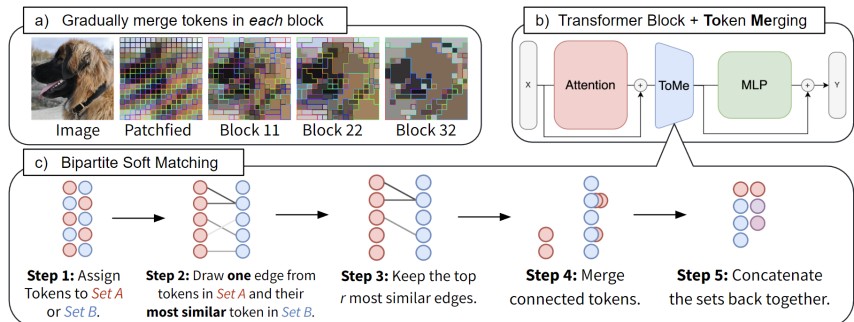

Figure 1: **Token Merging.** (a) With ToMe, similar patches are merged in each transformer block: for example, the dog's fur is merged into a single token. (b) ToMe is simple and can be inserted inside the standard transformer block. (c) Our fast merging algorithm, see Appendix D for implementation.

values of each token ($16 * 16 * 3 = 768$). This means that the intermediate features have the potential to contain insignificant noise that would confound our similarity calculations.

Luckily, transformers natively solve this problem with QKV self-attention (Vaswani et al., 2017). Specifically, the keys (K) already summarize the information contained in each token for use in dot product similarity. Thus, we use a dot product similarity metric (e.g., cosine similarity) between the keys of each token to determine which contain similar information (see Tab. 1a, 1b).

**Bipartite Soft Matching.** With token similarity defined, we need a *fast* way to determine which tokens to *match* in order to reduce the total number by $r$. There are several potential solutions to this problem, such as kmeans clustering (Lloyd, 1982) or graph cuts (Boykov et al., 2001). But we perform this matching $L$ times within the network on potentially thousands of tokens, so its runtime has to be *absolutely negligible*. This is very much not the case for most iterative clustering algorithms.

Thus, we propose a more efficient solution. Our design goals are as follows: 1.) we want to avoid anything iterative that cannot be parallelized and 2.) we want the changes merging makes to be *gradual*. The latter is why we focus on *matching* and not *clustering*, as clustering places no bounds on how many tokens can be merged into one group (which may adversely affect the network) , whereas matching leaves most of the tokens unmerged. Our algorithm is as follows (visualized in Fig. 1):

1. Partition the tokens into two sets $\mathbb{A}$ and $\mathbb{B}$ of roughly equal size.
2. Draw **one** edge from each token in $\mathbb{A}$ to its *most similar* token in $\mathbb{B}$.
3. Keep the $r$ most similar edges.
4. Merge tokens that are still connected (e.g., by averaging their features).
5. Concatenate the two sets back together.

Because this creates a bipartite graph and each token in $\mathbb{A}$ has only one edge, finding connected components in step 4 is trivial. Moreover, we don't need to compute similarity between every pair of tokens which, if we choose $\mathbb{A}$ and $\mathbb{B}$ carefully, isn't a problem for accuracy (see Tab. 1e). In fact, this "bipartite soft matching" is nearly as fast as just dropping tokens randomly (see Tab. 2) and takes only a few lines of code to implement (see Appendix D).

**Tracking Token Size.** Once tokens are merged, they no longer represent one input patch. This can change the outcome of softmax attention: if we merge two tokens with the same key, that key has less effect in the softmax term. We can fix this with a simple change, denoted *proportional attention*:

$$A = \text{softmax}\left(\frac{QK^\top}{\sqrt{d}} + \log s\right) \qquad (1)$$

where $s$ is a row vector containing the *size* of each token (number of patches the token represents). This performs the same operation as if you'd have $s$ copies of the key. We also need to weight tokens by $s$ any time they would be aggregated, like when merging tokens together (see Tab. 1d).

**Training with Merging.** Each component thus far has been designed to be able to add token merging to an already trained ViT model. Training with ToMe isn't necessary, but it may be desirable to

| feature | acc | im/s |
|---|---|---|
| $X_{pre}$ | 83.02 | **186.8** |
| X | 83.70 | 182.8 |
| K | **84.25** | 182.9 |
| Q | 84.04 | 182.8 |
| V | 83.80 | 182.9 |

(a) **Feature Choice.** The K matrix accurately summarizes the information within tokens.

| function | acc | im/s |
|---|---|---|
| eucl | **84.26** | 182.5 |
| cosine | **84.25** | **182.9** |
| dot | 82.78 | **183.0** |
| softmax | 82.00 | **183.0** |

(b) **Distance Function.** Cosine similarity is the best choice for speed and accuracy.

| aggregate | acc | im/s |
|---|---|---|
| concat | **84.32** | 180.3 |
| mean | 84.25 | **182.9** |

(c) **Head Aggregation.** Averaging over the attention heads is a bit less accurate, but faster.

| method | acc | im/s |
|---|---|---|
| keep one | 81.01 | **185.4** |
| max pool | 83.50 | 184.6 |
| avg pool | 83.57 | 183.8 |
| weighted avg | **84.25** | 182.9 |

(d) **Combining Method.** Averaging tokens weighted by their size, $s$ (see Eq. 1), ensures consistency.

| order | acc | im/s |
|---|---|---|
| sequential | 81.07 | **183.0** |
| alternating | 84.25 | 182.9 |
| random | 83.80 | 181.7 |

(e) **Partition Style.** Alternating $\mathbb{A}$ and $\mathbb{B}$ ensures tokens are compared in subsequent layers.

| src | prop | acc | im/s |
|---|---|---|---|
| mae | | **84.25** | **182.9** |
| mae | ✓ | 83.84 | 180.9 |
| augreg | | 82.15 | **182.8** |
| augreg | ✓ | **83.51** | 180.8 |

(f) **Proportional Attn.** Without MAE pretraining, off-the-shelf models require prop attn.

Table 1: **Token Merging ablation experiments** using ViT-L/16 from MAE (He et al., 2022) on ImageNet-1k evaluated off-the-shelf *without training*, using $r = 8$. The baseline model without ToMe obtains 85.96% acc at 93.3 im/s. For each ablation, we report Top-1 accuracy (acc) and fp32 model throughput (im/s) on a V100 GPU. Our default settings are marked in purple .

reduce accuracy drop or to speed up training. To train, we simply treat token merging as a pooling operation and backprop through the merged tokens as if we were using average pooling. We don't find a need to use any gradient tricks such as Gumbel softmax (Jang et al., 2017) as in token pruning (e.g., Kong et al. (2022)). And in fact, we find that the same settings used in training a vanilla ViT are also optimal here (see Appendix B). Thus ToMe is a drop-in replacement to increase training speed.

## 4    IMAGE EXPERIMENTS

We perform several experiments on ImageNet-1k (Deng et al., 2009) using ViT models trained in four different ways: AugReg (Steiner et al., 2022), MAE (He et al., 2022), SWAG (Singh et al., 2022), and DeiT (Touvron et al., 2021). For all experiments, we either run the model *off-the-shelf* with our method or, in the case of MAE and DeiT, *trained* with our method applied. All throughputs are measured during inference on a V100 GPU with optimal batch size and fp32 unless noted otherwise.

### 4.1    DESIGN CHOICES

In Tab. 1, we ablate the design choices made in our approach. For each ablation, we start from our default parameters marked in purple. Unless otherwise noted, we test on an off-the-shelf ViT-L/16 MAE model without training (acc: 85.96%, im/s: 93.3). and merge with $r = 8$ which gradually removes 98% of tokens over the 24 layers of the network.

**Token Similarity.** The tokens' features (X) are not the best in terms of performance (Tab. 1a). Moving the merging operation after attention (X vs. $X_{pre}$) and using the attention keys (K) is significantly more accurate. Then, cosine similarity is best to measure token distance as shown in Tab. 1b. Finally, we average K over the attention heads instead of concatenating them (Tab. 1c) for efficiency.

**Algorithmic Choices.** After deciding what tokens to merge, we combine them by averaging weighted by token size, $s$ (see Eq. 1). In Tab. 1d, this outperforms keeping just the token in $\mathbb{B}$, max pooling, or unweighted average pooling. Then, our bipartite matching algorithm requires splitting the input into two disjoint sets. Because we concatenate the sets afterward, we find that assigning tokens by alternating to work the best (Tab. 1e). Filling $\mathbb{A}$ and then filling $\mathbb{B}$ (sequentially) performs the worst.

**Proportional Attention.** Once merged, tokens can represent more than one input patch. We address this with proportional attention (Eq. 1), which we ablate in Tab. 1f. Surprisingly, proportional attention is necessary for supervised models (e.g., AugReg, SWAG, DeiT), but not for MAE models.

| style | algorithm | acc | im/s |
|-------|-----------|-----|------|
| prune | random | 79.22 | **184.4** |
| prune | attn-based | **79.48** | 183.8 |
| merge | kmeans (2 iter) | 80.19 | 169.7 |
| merge | kmeans (5 iter) | 80.29 | 147.5 |
| merge | greedy matching | **84.36** | 179.4 |
| merge | bipartite matching | 84.25 | **182.9** |

Table 2: **Matching Algorithm.** Different matching algorithms with the same settings as Tab. 1. Our bipartite algorithm is almost as fast as randomly pruning tokens, while retaining high accuracy. Matching is more suited for this setup than pruning or clustering.

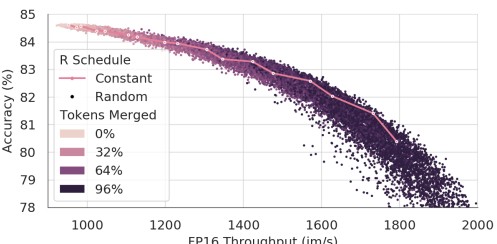

Figure 2: **Token Merging Schedule.** Our default constant merging schedule is close to optimal when compared to 15k randomly sampled merging schedules on an AugReg ViT-B/16.

Since this discrepancy disappears after training, this is likely because MAE already removes tokens during pretraining. Nevertheless, we use proportional attention for all but off-the-shelf MAE models.

**Comparing Matching Algorithms.** In Tab. 2 we compare our bipartite matching to different token reduction algorithms, both pruning and merging. Pruning is fast, but with 98% of the tokens removed overall, important information is lost. This is true for both pruning randomly and pruning based on what isn't attended to (Kim et al., 2021). In contrast, merging tokens only loses information when dissimilar tokens are merged. Thus, it's important to correctly select similar tokens to merge.

At first, kmeans (Lloyd, 1982) may seem like the obvious choice, but on top of being slow it's only slightly better than pruning. While it may minimize reconstruction error, kmeans allows a large number of tokens to match to the same cluster, which increases the probability of dissimilar tokens being merged. Marin et al. (2021) study several faster clustering algorithms based on kmeans, but they are unable to obtain a better than 10% accuracy drop in their setting without training.

Instead, we want a *matching* algorithm that only merges the most similar tokens. We could do this greedily by merging the most similar pair of tokens and then repeating without replacement $r$ times. This is accurate but sequential and thus could get slow with large $r$. Our bipartite matching has the accuracy of this greedy approach and the speed of pruning while having constant runtime w.r.t. to $r$.

**Selecting a Merging Schedule.** By default, we merge tokens with a *constant* schedule, i.e. $r$ per layer. To evaluate the optimality of this design we randomly sample a total of 15,000 merging schedules. For each schedule, we test its accuracy and fp16 throughput on ImageNet-1k val using an off-the-shelf AugReg ViT-B/16 model. In Fig. 2, we plot the results of this experiment and find that a constant schedule is close to optimal, especially as the total tokens merged increases. We further analyze the best random samples (see Appendix C) and find that a linearly decreasing schedule works well at throughputs up to ~3x. Thus, we also define a "decreasing" schedule that removes $2r$ tokens in the first layer and 0 tokens in the last layer, linearly interpolating for the rest. This also removes $rL$ tokens, but is faster because more are removed early:

$$\text{Constant Schedule} \qquad x \text{ per layer} \qquad \text{denoted } r_x \rightarrow \qquad (2)$$
$$\text{Decreasing Schedule} \qquad 2x \rightarrow 0 \text{ per layer} \qquad \text{denoted } r_x \searrow \qquad (3)$$

## 4.2 MODEL SWEEP

In Fig. 3, we apply our token merging method to 11 SotA off-the-shelf ViT models from various sources. For each model, we vary $r$ with a constant schedule to construct throughput vs. accuracy curves, starting with 0 (no merging baseline) to $r$ such that we run out of tokens to merge. We evaluate each model *off-the-shelf*. That is, by default *we don't train*; we just download the model and change a few lines of code. Models are evaluated on 224px images unless otherwise noted.

**Supervised Models.** Both AugReg (Steiner et al., 2022) and SWAG (Singh et al., 2022) are ViT models pretrained on a large supervised (or weakly supervised) pretraining dataset and fine-tuned on ImageNet-1k. AugReg covers optimal ViT training up until ViT-L/16, while SWAG pushes the limits of ImageNet by training huge models with massive image sizes. We apply our method *off-the-shelf* on AugReg models in Fig. 3a and SWAG models in Fig. 3b.

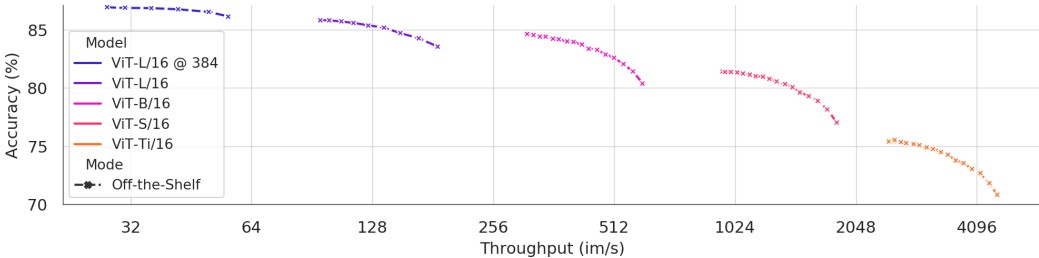

(a) **AugReg Models.** A collection of ImageNet-21k pretrained models (Steiner et al., 2022).

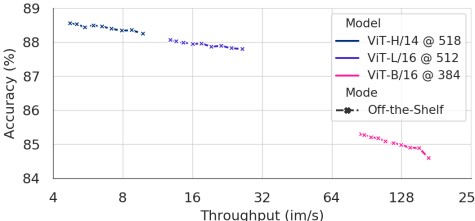

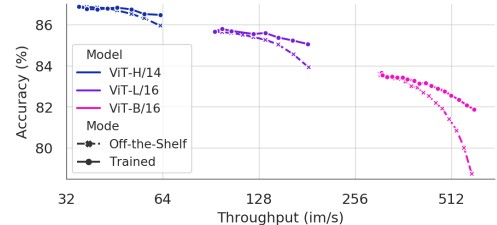

(b) **SWAG Models.** Massive weakly supervised models pretrained on 3.6B images (Singh et al., 2022).

(c) **MAE Models.** Self-supervised models pretrained on ImageNet-1k (He et al., 2022).

Figure 3: **Model Sweep.** We apply ToMe to several state of the art ViT models *off-the-shelf*, i.e. *without training*, varying $r$ to produce fp32 throughput vs. accuracy curves on ImageNet-1k.

Immediately, we can see that a constant schedule gives up to $2\times$ the throughput no matter the model. And even though we're compressing 96-98% of the tokens in each, the largest models have barely any accuracy drop: while ViT-B, S, and Ti all have around 4-5% accuracy drop at $2\times$ speed, ViT-L only suffers a 2% drop on 224px images and a 0.7% drop on 384px images with AugReg. Similarly, with SWAG models, ViT-L on 512px images and ViT-H on 518px images both have small 0.3% accuracy drop *without training*. Note this trend is not just because larger models have more tokens, since we always reduce the number of tokens by 96-98%. Instead, we think this is because large models are deeper and thus allow for more *gradual* change in features, which lessens the impact of merging.

**Self-Supervised Models.** MAE (He et al., 2022) is a self-supervised pretraining method for ViT with models pretrained and fine-tuned on ImageNet-1k. In Fig. 3c we apply our method both *off-the-shelf* and *trained* by fine-tuning the public pretrained checkpoints. When fine-tuning, we find that we can use the original training recipes. We don't have to to compensate for fewer tokens in later layers (see Appendix B), likely because our method is already tuned to imitate a model without merging.

And as expected, in Fig. 3c, we see the same trends as before: except this time, with training we can bring the error down to 0.4% for ViT-H, 0.6% for ViT-L, and 1.7% for ViT-B at $2\times$ throughput. Our approach actually implicitly addresses an issue in MAE: because MAE removes tokens during pretraining, its epoch times are $\sim4\times$ faster than training a supervised model. However, normal fine-tuning uses all the tokens and doesn't have this benefit. Our token merging method fixes this issue and allows for roughly $\sim2\times$ faster epochs at negligible accuracy drops for large models. This suggests that one could train even bigger models with token merging than was possible before.

**Re-evaluating.** Note that, while in Fig. 3c we train a new model for each value of $r$, this isn't actually necessary. Instead, we can take a model trained with one value of $r$ and re-evaluated it with another. In fact, it's possible to actually *improve* performance by doing so. For instance, the baseline ViT-L model we train in Fig. 3c gets 85.7% accuracy. If we re-evaluate our $r = 5$ trained model with $r = 0$, we obtain 85.8% accuracy. Thus, it's feasible to speed up training with ToMe and not apply it during evaluation to produce the same or better results. This means that, while the result of applying ToMe in Fig. 3b and Fig. 3c are similar to e.g. scaling the model size, you only have to train one model with ToMe to create any a model with a large range of scales.

### 4.3 COMPARISON TO OTHER WORKS

In this section, we compare our trained token merging models to other state-of-the art works on ImageNet-1k, both in terms of the overall vision space as well as other token reduction methods.

| model | input | acc | gflops | im/s |
|---|---|---|---|---|
| Eff-B5 | 456 | 83.6 | 9.9 | 169‡ |
| ViT-B $^{MAE}$ | 224 | 83.6 | 17.6 | 309 |
| Swin-B* | 224 | 84.0 | 15.4 | 278‡ |
| CSWin-B | 224 | 84.2 | 15.0 | 250‡ |
| MViTv2-B | 224 | 84.4 | 10.2 | 253‡ |
| **ToMe** | | | | |
| **ViT-L** $_{r_8\searrow}^{MAE}$ | 224 | 84.2 | 22.3 | 241 |
| **ViT-L** $_{r_8\rightarrow}^{MAE}$ | 224 | **85.1** | 31.0 | 183 |
| ViT-L $^{MAE}$ | 224 | 85.7† | 61.6 | 93 |
| Eff-B6 | 528 | 84.0 | 19.0 | 96‡ |
| MViTv2-L | 224 | 85.3 | 42.1 | 81 |
| **ToMe** | | | | |
| **ViT-H** $_{r_7\searrow}^{MAE}$ | 224 | 86.1 | 72.6 | 81 |
| **ViT-H** $_{r_7\rightarrow}^{MAE}$ | 224 | **86.5** | 92.9 | 63 |
| ViT-H $^{MAE}$ | 224 | 86.9† | 167.4 | 35 |
| SwinV2-H* | 224 | 85.7 | 118.1 | 49 |

Table 3: **Advancing a Tier.** Comparison to SoTA models trained only on ImageNet-1k. Our method allows for the use of more complicated models for the same tier of throughput. * models use SimMIM self-supervised pretraining. † baseline models trained by us without ToMe for comparison. ‡ im/s from original papers (V100).

| method | acc | inference | | train |
|---|---|---|---|---|
| | | gflops | im/s | speed |
| DeiT-S | 79.8 | 4.6 | 930 | 1× |
| A-ViT | 78.6† | 2.9 | - | 1× |
| DynamicViT | 79.3 | 2.9 | 1505 | 1× |
| SP-ViT | 79.3 | 2.6 | - | 1× |
| **ToMe** $_{r_{13}\rightarrow}^{DeiT}$ | **79.4** | 2.7 | **1552** | **1.5×** |
| **ToMe** $_{r_{13}\rightarrow}^{AugReg}$ | 79.3 | 2.7 | **1550** | - |

Table 4: **ToMe vs. pruning methods** on **ViT-S** trained from scratch with DeiT. We time DynamicViT on our V100, but the others cannot be batched and evaluate throughput in different settings. Pruning methods require token padding during training, and thus don't improve training speed, while ToMe at $r = 13$ trains 1.5× faster than the baseline DeiT-S. We also obtain the same result *without training* on an off-the-shelf AugReg ViT-S model. blue indicates ToMe applied *without training* while gray indicates ToMe applied during training. † A-ViT is not trained from scratch and performs slightly better than DynamicViT in its setting. See Appendix A for DeiT-Ti results.

**Comparison to State of the Art.** In Tab. 3, we compare our MAE fine-tuned models to state-of-the-art models trained on ImageNet-1k without extra data: EfficientNet (Tan & Le, 2019), Swin (Liu et al., 2021), SwinV2 (Liu et al., 2022a), CSWin (Dong et al., 2022), and MViTv2 (Li et al., 2022). All throughputs are on a single V100. Note that we use MAE pretraining which is not supported for all transformers, but provides accuracy improvement for some like Swin/SwinV2. Thus we also include SimMIM (Xie et al., 2022) pre-trained Swin and SwinV2 models for comparison.

Nevertheless, token merging with ToMe improves the throughput of ViT models such that ViT-L and ViT-H become comparable in speed to models of a lower tier, without scaling the number of features. Thus we display results of ViT "advancing a tier" in Tab. 3. More testing is need to see whether applying ToMe and model scaling at the same time would produce even better results.

**Comparison to Token Pruning.** In Tab. 4, we compare ToMe to token pruning works that use DeiT-S training[3]: A-ViT (Yin et al., 2022), DynamicViT (Rao et al., 2021), and SP-ViT (Kong et al., 2022) with throughput measured on a V100. Even though we don't use gradient tricks such as gumbel softmax (Jang et al., 2017), add extra parameters, or use additional training tricks, we can already match the performance and exceed the throughput of existing much more complicated token pruning works. Moreover, most token pruning works are forced to use padding tokens or attention masking during training, negating the benefits of pruning in the first place. Our method, on the other hand, doesn't suffer from this issue and we observe a 1.5× training speedup with DeiT. Interestingly, after 300 epochs the DeiT models have a similar accuracy drop to our MAE trained ViT-L (Appendix A). But we actually don't need to train at all: if we take an off-the-shelf AugReg ViT-S model and apply the same merging schedule, we can match the performance of the DeiT models *without training*.

## 4.4 VISUALIZATIONS

In Fig. 4, we show the input patches belonging to each merged token at the end of the network. We find that applying ToMe results in token merging that resembles something not unlike part segmentation (Chen et al., 2014). In the second image, the husky has different tokens for its legs, body, and face. The monkey in the 3rd image has different tokens for its hand, body, face, eyes, and mouth while the orange it's holding gets its own token despite it representing just one input

---

[3]This comparison is difficult as many token pruning works use different training strategies, some even claiming improvement in accuracy without a valid baseline. A-ViT fine-tunes on top of DeiT, while DynamicViT starts DeiT training from an existing checkpoint. We, on the other hand, train from scratch.

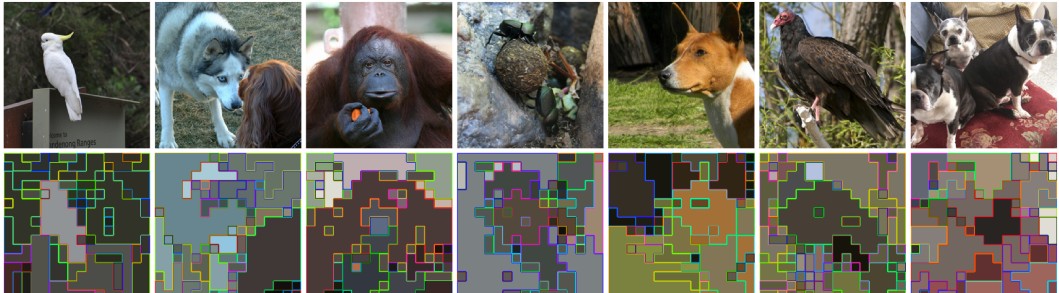

Figure 4: **Image Visualizations.** Results of merging on ImageNet-1k val using a ViT-H$_{r7 \rightarrow}^{\text{MAE}}$ model trained with ToMe. Patches with the same inner and border color are merged together. Unlike pruning, ToMe can merge similar parts of the image whether they're in the foreground or background.

patch. In cases where there are multiple instances of the same class like the dung beetles in the fourth image and the Boston terriers in the last image, the same parts from all instances get merged together. Notably unlike pruning, ToMe is able to merge a large number of tokens both in the background and the foreground without losing information. See more results and methodology in Appendix E.

## 5 VIDEO EXPERIMENTS

This framework of MAE plus token merging is a powerful strategy across several domains. Because of its high redundancy, one of the most promising directions is video. Thus, we apply our token merging approach on Spatiotemporal MAE (Feichtenhofer et al., 2022) for video classification on Kinetics-400 (Kay et al., 2017), both by simply applying our method off-the-shelf *without training* and by applying our method during MAE fine tuning with the default training recipe like we did for images. Note that nothing in our method needs to change for video: we use the same code for both.

**Results.** In Tab. 5, we show the results of applying our method *off-the-shelf* and during MAE fine-tuning using ViT-L from Spatiotemporal MAE compared to the relevant state-of-the-art on Kinetics-400 classification: Swin (Liu et al., 2022b) pretrained on ImageNet-21k, MViTv2 (Li et al., 2022) pretrained with MaskFeats (Wei et al., 2022), and Spatiotemporal MAE as the baseline. We also include a token pruning work, X-ViT + ATS (Fayyaz et al., 2022), for completeness.

Amazingly, ToMe applied to ViT-L with a constant schedule can match the throughput of Swin-B while performing better than MViTv2-L, even when evaluated *without training*. Moreover, with a decreasing schedule, ViT-L $_{r65 \searrow}^{\text{MAE}}$ significantly outperforms the baseline ViT-B$^{\text{MAE}}$ model with the same flop count with or without training, meaning ToMe is better than model scaling here. Training is not necessary for a constant schedule, but it does help with a decreasing schedule.

**Throughput.** In Tab. 6, we display the throughput and training time of our method applied to ViT-L. With a constant schedule, we can increase throughput by $2.2\times$ for a negligible 0.2% accuracy drop. Moreover, this setting *cuts training time in half*, even with the overhead of syncing across 8 gpus.

**Clip Count.** Because each forward pass only sees up to 2 seconds of video, it's standard practice to evaluate video recognition models with multiple clips. In Tab. 5, we evaluate with multiple clips (1 spatial crop, 10 temporal crops). We don't factor the number of clips into flop count because this is a hyperparameter every method can tune, usually resulting in only small differences as long as a minimum number of clips are used (i.e., 4 in this case). Thus, we choose the same number of clips as other models to compare. However, this might compensate for the information loss from token merging. In Fig. 5, we test if this is the case by sweeping over the number of clips for our method compared to the baseline ViT-L model. For $r = 65$, we see some degradation compared to the 4 clip sweet-spot ($\sim 0.5\%$), but for lower $r$ values, there's no decrease compared to the baseline.

**Visualization.** We visualize the final tokens for each input patch over multiple frames of video in Fig. 6 using our trained ViT-L $_{r65 \rightarrow}^{\text{MAE}}$ model. Just like ToMe performs primitive part segmentation on images, it is actually able to perform primitive part *tracking* on video. The same object or part is merged into one across multiple frames of video like the ball in Fig. 6. Note that the extraneous red patch in the third frame is the reflection of the ball in the glass. More results in Appendix E.

| model | input | acc | gflops |
|---|---|---|---|
| XViT + ATS | $16 \times 224^2$ | 80.0 | $259 \times 1 \times 3$ |
| ViT-B $^{MAE}$ | $16 \times 224^2$ | 81.3 | $\mathbf{180} \times 3 \times 7$ |
| Swin-B$^\dagger$ | $32 \times 224^2$ | 82.7 | $282 \times 3 \times 4$ |
| **ToMe** | | | |
| **ViT-L** $^{MAE}_{r_{65}\searrow}$ | $16 \times 224^2$ | 82.5 | $\mathbf{184} \times 1 \times 10$ |
| **ViT-L** $^{MAE}_{r_{65}\searrow}$ | $16 \times 224^2$ | **83.2** | $\mathbf{184} \times 1 \times 10$ |
| ViT-L $^{MAE}$ | $16 \times 224^2$ | 84.7 | $598 \times 1 \times 10$ |
| Swin-L$^\dagger$ | $32 \times 224^2$ | 83.1 | $604 \times 3 \times 4$ |
| MViTv2-L | $16 \times 224^2$ | 84.3 | $377 \times 1 \times 10$ |
| **ToMe** | | | |
| **ViT-L** $^{MAE}_{r_{55}\rightarrow}$ | $16 \times 224^2$ | **84.5** | $325 \times 1 \times 4$ |
| **ViT-L** $^{MAE}_{r_{65}\rightarrow}$ | $16 \times 224^2$ | **84.5** | $\mathbf{281} \times 1 \times 10$ |
| **ViT-L** $^{MAE}_{r_{65}\rightarrow}$ | $16 \times 224^2$ | 84.4 | $\mathbf{281} \times 1 \times 10$ |

Table 5: **Video Results.** Results for our method *without training* ( blue ) or applied during MAE fine-tuning ( gray ) compared to SoTA on Kinetics-400 in the same flop range. $\dagger$ uses ImageNet-21k pretraining while others only use Kinetics-400. Original baseline model in gray.

| model | clip/s | | ft time (8gpu) | |
|---|---|---|---|---|
| ViT-L $^{MAE}$ | 7.3 | $1.0\times$ | 263 hrs$^\ddagger$ | $1.0\times$ |
| **ViT-L** $^{MAE}_{r_{65}\rightarrow}$ | 16.3 | $2.2\times$ | 136 hrs | $0.5\times$ |
| **ViT-L** $^{MAE}_{r_{65}\searrow}$ | 24.9 | $3.4\times$ | - | - |

Table 6: **Video Throughput.** Inference speed and fine-tuning time for our method applied to video. ToMe *halves* training time for almost free. $^\ddagger$ estimated based on 5 days of training.

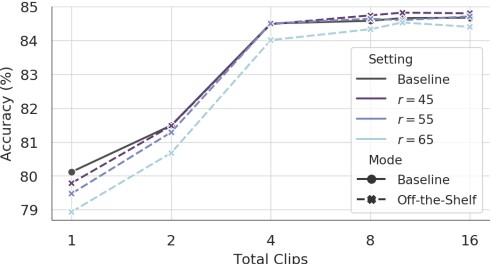

Figure 5: **Varying Clips.** ToMe closely matches the baseline's accuracy even with fewer clips.

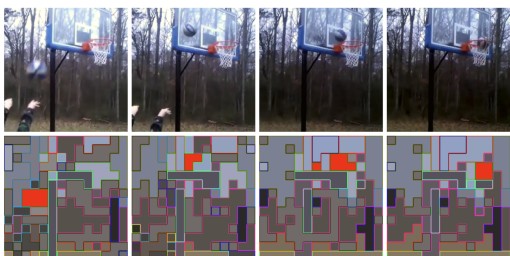

Figure 6: **Video Visualization.** ViT-L $^{MAE}_{r_{65}\rightarrow}$ with ToMe merges the same object across multiple frames of video. Here, the ball is merged into one token over its entire range of motion (red).

| model | mAP | gflops | sample/s |
|---|---|---|---|
| ViT-B$^{MAE}$ | 47.3 | 48.6 | 103 |
| **ViT-B**$^{MAE}_{r_{20}\rightarrow}$ | 46.2 | 36.3 | 136 |
| **ViT-B**$^{MAE}_{r_{40}\rightarrow}$ | 43.1 | 24.7 | **200** |
| ViT-B$^{MAE}$ | 46.4$^*$ | 48.6 | 103 |
| **ViT-B**$^{MAE}_{r_{20}\rightarrow}$ | 46.3 | 36.3 | 136 |
| **ViT-B**$^{MAE}_{r_{40}\rightarrow}$ | 46.0 | 24.7 | **200** |

Table 7: **Audio Results.** ViT-B fine-tuned from audio MAE pretraining on AudioSet-2M. ToMe can double the throughput of the baseline with only a 0.4% mAP drop. $^*$ due training implementation differences, the baseline we train performs worse than the original baseline.

# 6 AUDIO EXPERIMENTS

We perform experiments on an Audio MAE (Huang et al., 2022), where a spectrogram of the audio signal is rasterized and then fed into a ViT model. We use the ViT-B model from Huang et al. (2022) and evaluate on AudioSet-2M (Gemmeke et al., 2017). **Results.** Note, the metric reported is mAP instead of accuracy because of class imbalance. Due to training implementation differences, the baseline model we train has lower mAP than originally reported in Huang et al. (2022). Thus in Tab. 7, we compare ToMe without training to the original number, and ToMe with training to our trained baseline. Regardless, on audio we obtain an almost $2\times$ throughput increase with an mAP drop of only 0.4%. Full results for this experiment are in Appendix A.

# 7 CONCLUSION

In this work, we introduced Token Merging (ToMe) to increase the throughput of ViT models by gradually merging tokens. ToMe naturally exploits redundancy in the input, allowing its use for any modality with redundancy. In the paper we explore extensive experiments on images, video, and audio, obtaining speeds and accuracies competitive with the state-of-the-art in each case.

ToMe can be viewed as a "natural" hierarchical model, similar to Swin or MViT but using pure transformer blocks. ToMe could be combined with these methods to create an entirely new type of architecture. Similarly, we focus on classification but our visualizations show potential on tasks like segmentation. Finally, ToMe works well on large models across domains and cuts down training time and memory usage, meaning it could be a core component of training huge models. We leave these as topics for future work and hope ToMe can lead to the creation of better, more efficient transformers.

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

# A FULL RESULTS

Results for plots and tables in the main paper. For all results, im/s indicates throughput and "speed" indicates improvement over the baseline. All throughputs are measured on a V100, but the actual values may differ a little from the main paper as the model may have been benchmarked on a different machine. However, all results in the main paper use the same machine for throughput evaluation.

## A.1 IMAGES

For each ImageNet-1k model, we display our full results here.

### A.1.1 AUGREG MODELS

Full results listed in Tab. 8. We make no special changes to any of these models.

| model | r | acc | drop | im/s | speed | model | r | acc | drop | im/s | speed |
|---|---|---|---|---|---|---|---|---|---|---|---|
| ViT-B/16 | 0 | 84.57 | 0.00 | 309 | 1.00 | ViT-S/16 | 0 | 81.41 | 0.00 | 953 | 1.00 |
| ViT-B/16 | 1 | 84.61 | 0.03 | 311 | 1.01 | ViT-S/16 | 1 | 81.37 | -0.04 | 967 | 1.01 |
| ViT-B/16 | 2 | 84.52 | -0.05 | 322 | 1.04 | ViT-S/16 | 2 | 81.35 | -0.06 | 1001 | 1.05 |
| ViT-B/16 | 3 | 84.39 | -0.18 | 334 | 1.08 | ViT-S/16 | 3 | 81.30 | -0.10 | 1036 | 1.09 |
| ViT-B/16 | 4 | 84.39 | -0.18 | 346 | 1.12 | ViT-S/16 | 4 | 81.24 | -0.17 | 1072 | 1.12 |
| ViT-B/16 | 5 | 84.24 | -0.33 | 360 | 1.17 | ViT-S/16 | 5 | 81.12 | -0.29 | 1114 | 1.17 |
| ViT-B/16 | 6 | 84.17 | -0.40 | 373 | 1.21 | ViT-S/16 | 6 | 81.02 | -0.38 | 1151 | 1.21 |
| ViT-B/16 | 7 | 83.99 | -0.58 | 391 | 1.27 | ViT-S/16 | 7 | 80.94 | -0.46 | 1202 | 1.26 |
| ViT-B/16 | 8 | 83.94 | -0.63 | 406 | 1.31 | ViT-S/16 | 8 | 80.78 | -0.63 | 1247 | 1.31 |
| ViT-B/16 | 9 | 83.73 | -0.84 | 425 | 1.37 | ViT-S/16 | 9 | 80.53 | -0.88 | 1302 | 1.37 |
| ViT-B/16 | 10 | 83.37 | -1.21 | 443 | 1.44 | ViT-S/16 | 10 | 80.33 | -1.08 | 1364 | 1.43 |
| ViT-B/16 | 11 | 83.28 | -1.30 | 464 | 1.50 | ViT-S/16 | 11 | 80.06 | -1.35 | 1424 | 1.49 |
| ViT-B/16 | 12 | 82.86 | -1.72 | 488 | 1.58 | ViT-S/16 | 12 | 79.60 | -1.81 | 1487 | 1.56 |
| ViT-B/16 | 13 | 82.60 | -1.98 | 511 | 1.65 | ViT-S/16 | 13 | 79.30 | -2.10 | 1564 | 1.64 |
| ViT-B/16 | 14 | 82.04 | -2.54 | 540 | 1.75 | ViT-S/16 | 14 | 78.89 | -2.52 | 1646 | 1.73 |
| ViT-B/16 | 15 | 81.39 | -3.18 | 571 | 1.85 | ViT-S/16 | 15 | 78.14 | -3.26 | 1738 | 1.82 |
| ViT-B/16 | 16 | 80.38 | -4.20 | 603 | 1.95 | ViT-S/16 | 16 | 77.01 | -4.40 | 1836 | 1.93 |
| | | | | | | | | | | | |
| ViT-L/16 | 0 | 85.82 | 0.00 | 95 | 1.00 | ViT-Ti/16 | 0 | 75.50 | 0.00 | 2558 | 1.00 |
| ViT-L/16 | 1 | 85.80 | -0.02 | 100 | 1.05 | ViT-Ti/16 | 1 | 75.39 | -0.10 | 2471 | 0.97 |
| ViT-L/16 | 2 | 85.70 | -0.12 | 107 | 1.13 | ViT-Ti/16 | 2 | 75.40 | -0.09 | 2551 | 1.00 |
| ViT-L/16 | 3 | 85.58 | -0.24 | 115 | 1.22 | ViT-Ti/16 | 3 | 75.34 | -0.15 | 2651 | 1.04 |
| ViT-L/16 | 4 | 85.37 | -0.45 | 125 | 1.32 | ViT-Ti/16 | 4 | 75.27 | -0.23 | 2735 | 1.07 |
| ViT-L/16 | 5 | 85.17 | -0.65 | 137 | 1.44 | ViT-Ti/16 | 5 | 75.18 | -0.32 | 2849 | 1.11 |
| ViT-L/16 | 6 | 84.71 | -1.11 | 150 | 1.58 | ViT-Ti/16 | 6 | 75.06 | -0.44 | 2944 | 1.15 |
| ViT-L/16 | 7 | 84.26 | -1.56 | 167 | 1.76 | ViT-Ti/16 | 7 | 74.88 | -0.62 | 3074 | 1.20 |
| ViT-L/16 | 8 | 83.55 | -2.27 | 186 | 1.96 | ViT-Ti/16 | 8 | 74.76 | -0.74 | 3191 | 1.25 |
| | | | | | | ViT-Ti/16 | 9 | 74.50 | -1.00 | 3331 | 1.30 |
| ViT-L/16@384 | 0 | 86.92 | 0.00 | 28 | 1.00 | ViT-Ti/16 | 10 | 74.26 | -1.24 | 3469 | 1.36 |
| ViT-L/16@384 | 5 | 86.87 | -0.04 | 31 | 1.12 | ViT-Ti/16 | 11 | 73.76 | -1.73 | 3629 | 1.42 |
| ViT-L/16@384 | 10 | 86.85 | -0.07 | 36 | 1.28 | ViT-Ti/16 | 12 | 73.53 | -1.97 | 3792 | 1.48 |
| ViT-L/16@384 | 15 | 86.75 | -0.17 | 42 | 1.50 | ViT-Ti/16 | 13 | 73.04 | -2.46 | 3980 | 1.56 |
| ViT-L/16@384 | 20 | 86.53 | -0.39 | 50 | 1.78 | ViT-Ti/16 | 14 | 72.65 | -2.84 | 4175 | 1.63 |
| ViT-L/16@384 | 23 | 86.14 | -0.78 | 56 | 2.00 | ViT-Ti/16 | 15 | 71.80 | -3.70 | 4396 | 1.72 |
| | | | | | | ViT-Ti/16 | 16 | 70.79 | -4.71 | 4610 | 1.80 |

<table>
<tr><td>(a) ViT-B, ViT-L, and ViT-L at 384px.</td><td>(b) ViT-S and ViT-Ti.</td></tr>
</table>

Table 8: **Full AugReg Off-the-Shelf Results**. Results are *without training*. The original off-the-shelf models are listed in gray. We include the  blue  model in Tab. 4.

### A.1.2 SWAG MODELS

Full results listed in Tab. 9. Again, we make no special changes for these models.

| model | $r$ | acc | drop | im/s | speed |
|---|---|---|---|---|---|
| ViT-B/16 @ 384 | 0 | 85.30 | 0.00 | 85.7 | 1.00 |
| ViT-B/16 @ 384 | 5 | 85.27 | -0.03 | 88.6 | 1.03 |
| ViT-B/16 @ 384 | 10 | 85.21 | -0.09 | 94.6 | 1.10 |
| ViT-B/16 @ 384 | 15 | 85.18 | -0.13 | 101.7 | 1.19 |
| ViT-B/16 @ 384 | 20 | 85.09 | -0.22 | 109.1 | 1.27 |
| ViT-B/16 @ 384 | 25 | 85.03 | -0.27 | 118.3 | 1.38 |
| ViT-B/16 @ 384 | 30 | 84.98 | -0.33 | 127.7 | 1.49 |
| ViT-B/16 @ 384 | 35 | 84.90 | -0.40 | 139.6 | 1.63 |
| ViT-B/16 @ 384 | 40 | 84.89 | -0.41 | 152.8 | 1.78 |
| ViT-B/16 @ 384 | 45 | 84.59 | -0.71 | 167.7 | 1.96 |
| ViT-L/16 @ 512 | 0 | 88.06 | 0.00 | 12.8 | 1.00 |
| ViT-L/16 @ 512 | 5 | 88.02 | -0.04 | 13.7 | 1.06 |
| ViT-L/16 @ 512 | 10 | 87.98 | -0.08 | 14.7 | 1.14 |
| ViT-L/16 @ 512 | 15 | 87.95 | -0.11 | 16.1 | 1.25 |
| ViT-L/16 @ 512 | 20 | 87.96 | -0.10 | 17.5 | 1.36 |
| ViT-L/16 @ 512 | 25 | 87.87 | -0.19 | 19.3 | 1.51 |
| ViT-L/16 @ 512 | 30 | 87.89 | -0.17 | 21.3 | 1.66 |
| ViT-L/16 @ 512 | 35 | 87.82 | -0.24 | 23.7 | 1.84 |
| ViT-L/16 @ 512 | 40 | 87.80 | -0.26 | 26.3 | 2.05 |
| ViT-H/14 @ 518 | 0 | 88.55 | 0.00 | 4.7 | 1.00 |
| ViT-H/14 @ 518 | 5 | 88.53 | -0.02 | 5.1 | 1.07 |
| ViT-H/14 @ 518 | 10 | 88.44 | -0.11 | 5.5 | 1.16 |
| ViT-H/14 @ 518 | 15 | 88.49 | -0.06 | 6.0 | 1.26 |
| ViT-H/14 @ 518 | 20 | 88.46 | -0.09 | 6.5 | 1.38 |
| ViT-H/14 @ 518 | 25 | 88.39 | -0.16 | 7.2 | 1.51 |
| ViT-H/14 @ 518 | 30 | 88.34 | -0.21 | 7.9 | 1.67 |
| ViT-H/14 @ 518 | 35 | 88.35 | -0.19 | 8.8 | 1.85 |
| ViT-H/14 @ 518 | 40 | 88.25 | -0.30 | 9.8 | 2.06 |

Table 9: **Full SWAG Off-the-Shelf Results**. Results are *without training*. The original off-the-shelf models are listed in gray. We mention the blue models in the abstract.

### A.1.3 MAE MODELS

We evaluate MAE models both off the shelf and trained with ToMe in Tab. 10. For off-the-shelf evaluation we disable proportional attention as noted in Sec. 4.1, but we enable it for the trained models. Note that we compare to baselines we trained ourselves, which may slightly underperform the official baselines (for ViT-L). When training, we fine-tune from the official pretrained weights and use the original training recipe. Unlike prior work, we intend for ToMe to *replace* standard training, not augment it, in order to receive the benefits of faster training times and less memory usage.

### A.1.4 DEIT MODELS

We present DeiT results in Tab. 11. For DeiT, we train from scratch with the default training recipe for 300 epochs. Unlike other token pruning works, we don't use any tricks such as starting from an existing checkpoint or fine-tuning. Note that for merging, in addition to not merging the class token, we don't merge the distillation token. In Tab. 11, we don't train for all values of $r$, just the baseline $r = 0$ and those between 8 and 16. $r = 11$ for DeiT-S didn't finish training.

### A.2 VIDEO

We run the ViT-L model from Feichtenhofer et al. (2022) off the shelf. In Tab. 12, we show the results of this experiment by sweeping over $r$. For each setting, we evaluate with 1 spatial crop and 10 temporal clips. Note that the original baseline is evaluated with 3 spatial crops and 7 temporal clips, while we re-evaluated it with $1 \times 10$. Thus, the baseline has slightly lower accuracy than the original paper. Like with images, for these off-the-shelf MAE pretrained models we don't use proportional attention.

| model | $r$ | acc | drop | im/s | speed | | model | $r$ | acc | drop | im/s | speed |
|---|---|---|---|---|---|---|---|---|---|---|---|---|
| ViT-B/16 | 0 | 83.62 | 0.00 | 305 | 1.00 | | ViT-B/16 | 0 | 83.62 | 0.00 | 309 | 1.00 |
| ViT-B/16 | 1 | 83.55 | -0.07 | 307 | 1.01 | | ViT-B/16 | 1 | 83.60 | -0.02 | 311 | 1.01 |
| ViT-B/16 | 2 | 83.50 | -0.12 | 317 | 1.04 | | ViT-B/16 | 2 | 83.52 | -0.10 | 322 | 1.04 |
| ViT-B/16 | 3 | 83.44 | -0.18 | 329 | 1.08 | | ViT-B/16 | 3 | 83.49 | -0.13 | 334 | 1.08 |
| ViT-B/16 | 4 | 83.39 | -0.23 | 341 | 1.12 | | ViT-B/16 | 4 | 83.43 | -0.19 | 346 | 1.12 |
| ViT-B/16 | 5 | 83.36 | -0.26 | 355 | 1.16 | | ViT-B/16 | 5 | 83.43 | -0.20 | 360 | 1.17 |
| ViT-B/16 | 6 | 83.22 | -0.40 | 368 | 1.21 | | ViT-B/16 | 6 | 83.39 | -0.24 | 373 | 1.21 |
| ViT-B/16 | 7 | 83.01 | -0.61 | 384 | 1.26 | | ViT-B/16 | 7 | 83.31 | -0.31 | 391 | 1.27 |
| ViT-B/16 | 8 | 82.93 | -0.69 | 400 | 1.31 | | ViT-B/16 | 8 | 83.16 | -0.46 | 406 | 1.31 |
| ViT-B/16 | 9 | 82.69 | -0.93 | 418 | 1.37 | | ViT-B/16 | 9 | 83.20 | -0.42 | 425 | 1.37 |
| ViT-B/16 | 10 | 82.52 | -1.10 | 436 | 1.43 | | ViT-B/16 | 10 | 83.01 | -0.61 | 443 | 1.44 |
| ViT-B/16 | 11 | 82.18 | -1.44 | 457 | 1.50 | | ViT-B/16 | 11 | 82.94 | -0.68 | 464 | 1.50 |
| ViT-B/16 | 12 | 81.92 | -1.70 | 479 | 1.57 | | ViT-B/16 | 12 | 82.65 | -0.97 | 488 | 1.58 |
| ViT-B/16 | 13 | 81.41 | -2.21 | 504 | 1.65 | | ViT-B/16 | 13 | 82.55 | -1.07 | 511 | 1.65 |
| ViT-B/16 | 14 | 80.85 | -2.77 | 532 | 1.74 | | ViT-B/16 | 14 | 82.35 | -1.28 | 540 | 1.75 |
| ViT-B/16 | 15 | 80.01 | -3.61 | 561 | 1.84 | | ViT-B/16 | 15 | 82.12 | -1.50 | 571 | 1.85 |
| ViT-B/16 | 16 | 78.75 | -4.87 | 592 | 1.94 | | ViT-B/16 | 16 | 81.91 | -1.71 | 603 | 1.95 |
| | | | | | | | | | | | | |
| ViT-L/16 | 0 | 85.66 | 0.00 | 93 | 1.00 | | ViT-L/16 | 0 | 85.66 | 0.00 | 93 | 1.00 |
| ViT-L/16 | 1 | 85.63 | -0.03 | 98 | 1.06 | | ViT-L/16 | 1 | 85.79 | 0.13 | 98 | 1.05 |
| ViT-L/16 | 2 | 85.59 | -0.07 | 105 | 1.13 | | ViT-L/16 | 2 | 85.69 | 0.03 | 105 | 1.12 |
| ViT-L/16 | 3 | 85.51 | -0.15 | 114 | 1.23 | | | | | | | |
| ViT-L/16 | 4 | 85.39 | -0.27 | 123 | 1.33 | | ViT-L/16 | 4 | 85.54 | -0.12 | 123 | 1.32 |
| ViT-L/16 | 5 | 85.26 | -0.40 | 134 | 1.45 | | ViT-L/16 | 5 | 85.59 | -0.07 | 134 | 1.44 |
| ViT-L/16 | 6 | 85.03 | -0.63 | 147 | 1.59 | | ViT-L/16 | 6 | 85.37 | -0.28 | 147 | 1.58 |
| ViT-L/16 | 7 | 84.55 | -1.11 | 164 | 1.76 | | ViT-L/16 | 7 | 85.23 | -0.43 | 163 | 1.75 |
| ViT-L/16 | 8 | 83.92 | -1.74 | 183 | 1.97 | | ViT-L/16 | 8 | 85.05 | -0.61 | 183 | 1.96 |
| | | | | | | | ViT-L/16 | 8↘ | 84.16 | -1.50 | 241 | 2.58 |
| | | | | | | | | | | | | |
| ViT-H/14 | 0 | 86.88 | 0.00 | 35 | 1.00 | | ViT-H/14 | 0 | 86.88 | 0.00 | 35 | 1.00 |
| ViT-H/14 | 1 | 86.86 | -0.02 | 37 | 1.07 | | ViT-H/14 | 1 | 86.76 | -0.12 | 37 | 1.07 |
| ViT-H/14 | 2 | 86.82 | -0.06 | 40 | 1.14 | | ViT-H/14 | 2 | 86.73 | -0.15 | 40 | 1.14 |
| ViT-H/14 | 3 | 86.80 | -0.08 | 43 | 1.24 | | ViT-H/14 | 3 | 86.79 | -0.09 | 43 | 1.24 |
| ViT-H/14 | 4 | 86.69 | -0.19 | 46 | 1.34 | | ViT-H/14 | 4 | 86.82 | -0.06 | 46 | 1.34 |
| ViT-H/14 | 5 | 86.52 | -0.36 | 51 | 1.47 | | ViT-H/14 | 5 | 86.74 | -0.14 | 51 | 1.47 |
| ViT-H/14 | 6 | 86.31 | -0.58 | 56 | 1.62 | | ViT-H/14 | 6 | 86.51 | -0.37 | 56 | 1.62 |
| ViT-H/14 | 7 | 85.94 | -0.94 | 63 | 1.81 | | ViT-H/14 | 7 | 86.47 | -0.41 | 63 | 1.81 |
| | | | | | | | ViT-H/14 | 7↘ | 86.06 | -0.82 | 81 | 2.34 |

(a) **Applied *without* training.**          (b) **Applied during MAE fine-tuning.**

Table 10: **Full MAE Results**. Results are with and without training. The original baseline models we train are listed in gray. We include the ⬚gray⬚ models in Tab. 3.

## A.3 AUDIO

Full results for our audio experiments can be found in Tab. 13. We used the model from Huang et al. (2022) to evaluate off-the-shelf. However, for training we train with our own implementation that's different from the paper. For this reason, in Tab. 13, we list two different baselines (one from the original paper, and the other trained by us). In this case, we don't use proportional attention during off-the-shelf evaluation or training.

## B HYPERPARAMETERS

In Tab. 14, we perform a limited hyperparameter search on parameters that would be affected by applying ToMe: layer decay, drop path, and the number of epochs.

Layer decay reduces learning rate based on the layer of the network. Since ToMe gradually reduces the number of tokens, the size of gradient updates in later layers might already be lower without layer decay. However, we find that it's not necessary to change that parameter.

| $r$ | acc | drop | gflops | speedup |
|---|---|---|---|---|
| 0 | 79.96 | - | 4.61 | - |
| 8 | 79.68 | -0.28 | 3.43 | 1.34x |
| 9 | 79.69 | -0.27 | 3.28 | 1.40x |
| 10 | 79.59 | -0.37 | 3.14 | 1.47x |
| | | | | |
| 12 | 79.49 | -0.47 | 2.85 | 1.62x |
| 13 | 79.42 | -0.54 | 2.71 | 1.70x |
| 14 | 79.24 | -0.72 | 2.57 | 1.79x |
| 15 | 79.21 | -0.75 | 2.43 | 1.90x |
| 16 | 79.13 | -0.83 | 2.30 | 2.00x |

(a) **DeiT-S**. We include the gray result to compare against token pruning methods, though other speed-accuracy trade-offs may be more desirable. Note the baseline we train is slightly more accurate than the original.

| $r$ | acc | drop | gflops | speedup |
|---|---|---|---|---|
| 0 | 71.84 | - | 1.26 | - |
| 8 | 71.69 | -0.15 | 0.93 | 1.35x |
| 9 | 71.63 | -0.21 | 0.89 | 1.41x |
| 10 | 71.24 | -0.59 | 0.85 | 1.47x |
| 11 | 71.40 | -0.44 | 0.81 | 1.55x |
| 12 | 71.35 | -0.48 | 0.78 | 1.62x |
| 13 | 71.27 | -0.57 | 0.74 | 1.71x |
| 14 | 71.16 | -0.67 | 0.70 | 1.80x |
| 15 | 71.02 | -0.82 | 0.66 | 1.90x |
| 16 | 70.74 | -1.09 | 0.63 | 2.01x |

(b) **DeiT-Ti**. We omit DeiT-Ti results from the main paper because we are not able to reproduce the baseline accuracy in SPViT (Kong et al., 2022).

Table 11: **Full DeiT Results**. Results from running a standard 300 epoch DeiT-S and DeiT-Ti training run with the official codebase on ImageNet-1k. Note that we make no changes to hyperparameters or the model other than to add token merging. Not only is our method much simpler than other token reduction approaches, but this speed-up is observed both during inference and training. Speed-up and accuracy drop is relative to the baseline (gray).

| model | $r$ | top-1 | top-5 | gflops | clips/s | speed |
|---|---|---|---|---|---|---|
| ViT-L 16x4 | 0 | 84.66 | 96.47 | 598 | 7.3 | 1.00 |
| ViT-L 16x4 | 10 | 84.70 | 96.42 | 545 | 8.0 | 1.10 |
| ViT-L 16x4 | 20 | 84.82 | 96.43 | 493 | 9.0 | 1.23 |
| ViT-L 16x4 | 30 | 84.82 | 96.40 | 443 | 10.1 | 1.38 |
| ViT-L 16x4 | 40 | 84.86 | 96.40 | 394 | 11.4 | 1.56 |
| ViT-L 16x4 | 45 | 84.82 | 96.43 | 371 | 12.3 | 1.68 |
| ViT-L 16x4 | 50 | 84.69 | 96.45 | 348 | 13.1 | 1.79 |
| ViT-L 16x4 | 55 | 84.60 | 96.32 | 325 | 14.1 | 1.92 |
| ViT-L 16x4 | 60 | 84.51 | 96.30 | 303 | 15.1 | 2.06 |
| ViT-L 16x4 | 64 | 84.41 | 96.33 | 286 | 16.0 | 2.19 |
| ViT-L 16x4 | 65 | 84.51 | 96.33 | 281 | 16.3 | 2.22 |
| ViT-L 16x4 | 65↘ | 82.49 | 95.65 | 184 | 24.9 | 3.39 |

Table 12: **Full Video Off-the-Shelf Results**. Results are *without training*. The original model is listed in gray. We include the blue models in Tab. 5. Evaluation is $1 \times 10$ (this is not factored into the flop count). We include both top-1 and top-5 accuracy on Kinetics-400.

| r | mAP | drop | sample/s | speed |
|---|---|---|---|---|
| 0 | 47.29 | 0.00 | 103.3 | 1.00 |
| 5 | 47.16 | -0.13 | 108.7 | 1.05 |
| 10 | 46.92 | -0.37 | 115.6 | 1.12 |
| 15 | 46.60 | -0.69 | 125.8 | 1.22 |
| 20 | 46.21 | -1.08 | 136.2 | 1.32 |
| 25 | 45.70 | -1.59 | 148.9 | 1.44 |
| 30 | 45.00 | -2.29 | 162.7 | 1.58 |
| 35 | 44.18 | -3.11 | 181.3 | 1.76 |
| 40 | 43.09 | -4.20 | 199.9 | 1.94 |

(a) **ViT-B on AudioSet-2M *without* training.**

| r | mAP | drop | sample/s | speed |
|---|---|---|---|---|
| 0 | 46.43 | 0.00 | 103.3 | 1.00 |
| 5 | 46.21 | -0.23 | 108.7 | 1.05 |
| 10 | 46.36 | -0.08 | 115.6 | 1.12 |
| 15 | 46.17 | -0.27 | 125.8 | 1.22 |
| 20 | 46.29 | -0.14 | 136.2 | 1.32 |
| 25 | 46.09 | -0.34 | 148.9 | 1.44 |
| 30 | 46.09 | -0.34 | 162.7 | 1.58 |
| 35 | 46.12 | -0.31 | 181.3 | 1.76 |
| 40 | 45.96 | -0.47 | 199.9 | 1.94 |

(b) **ViT-B on AudioSet-2M with training.**

Table 13: **Full Audio Results**. Results are with and without training. The original baseline model (left) and the baseline we train (right) are listed in gray. We include the blue and gray models in Tab. 7.

Drop path randomly drops out entire attention and MLP blocks with some probability. This has the effect of regularizing layers so that they don't rely on a single block. Because we use the K matrix

from blocks that could be dropped out, we test the value of this parameter. Again, we find this not necessary to change.

We also perform the same experiments on video except with just layer decay and the number of epochs, testing whether ToMe requires increasing the number of epochs (due to seeing fewer tokens overall). And again, the default parameters work the best.

| layer decay | drop path | acc |
|---|---|---|
| 0.55 | 0.05 | 81.59 |
| 0.55 | 0.20 | 81.31 |
| 0.65 | 0.05 | 81.73 |
| 0.65 | 0.10 | **81.83** |
| 0.65 | 0.20 | 81.76 |
| 0.75 | 0.05 | 81.68 |
| 0.75 | 0.10 | 81.77 |
| 0.75 | 0.20 | 81.82 |
| 0.85 | 0.05 | 81.35 |
| 0.85 | 0.10 | 81.54 |
| 0.85 | 0.20 | 81.46 |

| layer decay | epochs | acc |
|---|---|---|
| 0.875 | 50 | 82.83 |
| 0.825 | 50 | 83.52 |
| 0.750 | 50 | **83.57** |
| 0.750 | 75 | 83.35 |
| 0.750 | 100 | 82.33 |

(a) **Image Fine-Tuning**. We sweep over ViT-B/16 MAE fine-tuning on ImageNet-1k with $r = 16$ and find that the default parameters from the official code release work the best.

(b) **Video Fine-Tuning**. We test some additional parameters for ViT-L spatiotemporal MAE fine-tuning on K400 with $r = 65$. The defaults also work best here. Accuracy is for 3x5 evaluation. Note that this fine-tuning started from a MAE pre-trained model trained for half its schedule, so these numbers aren't comparable to the numbers in Tab. 5.

Table 14: **Training Hyperparameters** don't need to be updated when training with token merging. We perform a sweep over relevant hyperparmaters that might be affected by token merging for images and video. The default setting, marked in purple , already has the highest accuracy.

## C   MERGING SCHEDULE

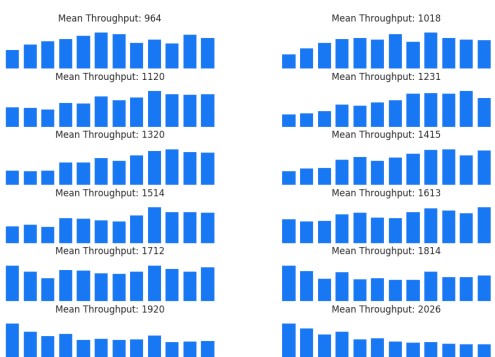

Figure 7: **Merging Schedule.** The average of the top 100 merging schedules for the experiment in Fig. 2. For each throughput range, we find the highest accuracy schedules and average their number of tokens merged per layer. For lower throughputs, merging more tokens at the end is better. For higher throughputs, a constant schedule becomes the best. Then for even higher throughputs, a linearly decreasing schedule works well.

In Fig. 7, we plot the average number of tokens merged in each layer for the most accurate random samples in Fig. 2. Around throughputs of 1600-1800, the best schedule is close to constant, which is why constant is close to optimal in this range. For throughputs beyond that, however, a decreasing schedule is best. For this, reason we define a linearly decreasing schedule in addition to a constant schedule in the main paper.

## D  IMPLEMENTATION

The following is an implementation of our "bipartite soft matching" in PyTorch (Paszke et al., 2019):

```python
def bipartite_soft_matching(k: torch.Tensor, r: int) -> torch.Tensor:
    """ Input is k from attention, size [batch, tokens, channels]. """
    k = k / k.norm(dim=-1, keepdim=True)
    a, b = k[..., ::2, :], k[..., 1::2, :]
    scores = a @ b.transpose(-1, -2)

    scores[..., 0, :] = -math.inf  # don't merge cls token

    node_max, node_idx = scores.max(dim=-1)
    edge_idx = node_max.argsort(dim=-1, descending=True)[..., None]

    unm_idx = edge_idx[..., r:, :] # Unmerged Tokens
    src_idx = edge_idx[..., :r, :] # Merged Tokens
    dst_idx = node_idx[..., None].gather(dim=-2, index=src_idx)

    unm_idx = unm_idx.sort(dim=-2)[0] # Sort cls token back to idx 0

    def merge(x: torch.Tensor) -> torch.Tensor:
        """ Input is of shape [batch, tokens, channels]. """
        src, dst = x[..., ::2, :], x[..., 1::2, :]
        n, t1, c = src.shape

        unm = src.gather(dim=-2, index=unm_idx.expand(n, t1 - r, c))
        src = src.gather(dim=-2, index=src_idx.expand(n, r, c))
        dst = dst.scatter_add(-2, dst_idx.expand(n, r, c), src)

        return torch.cat([unm, dst], dim=-2)

    return merge
```

This returns a lambda function that can be applied to any matrix or vector (i.e. to merge features, to calculate token size, or to calculate source patches). Note how this is done all at once in parallel—there are no sequential loops.

## E  MORE VISUALIZATION

To create the visualizations in Fig. 4 and Fig. 6, we follow each final merged token back to its original input patches. Then for each token, we color its input patches with the average color in that region. To make sure different tokens are distinct from each other, we also assign each token a random border color. Note that tokens do not necessarily represent contiguous input regions. The only spatial signal ToMe has comes from the position encodings.

In Fig. 8, we present several more examples of merging on images as a continuation of Fig. 4. ToMe's propensity for part and object segmentation appears time and time again across many different images.

In Fig. 9, we also display more results of ToMe performing object tracking on video. Note that in (Feichtenhofer et al., 2022), each token represents more than one frame. Namely, the patch size is $2 \times 16 \times 16$ and thus 2 frames of video correspond to each token. We plot the first frame from the two, because we find that more closely matches the merged tokens.

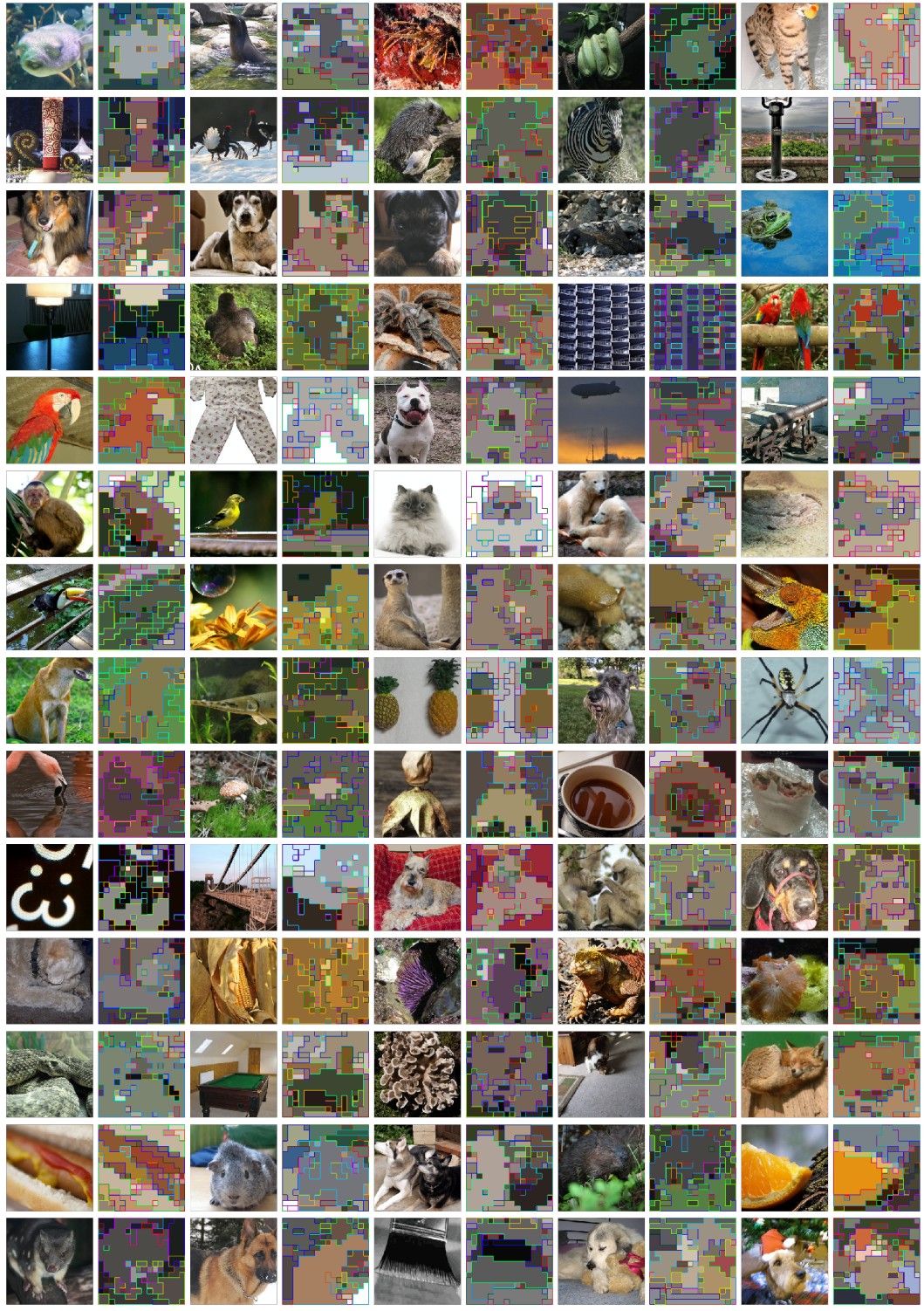

Figure 8: **More visualization on images.** Continuation of Fig. 4.

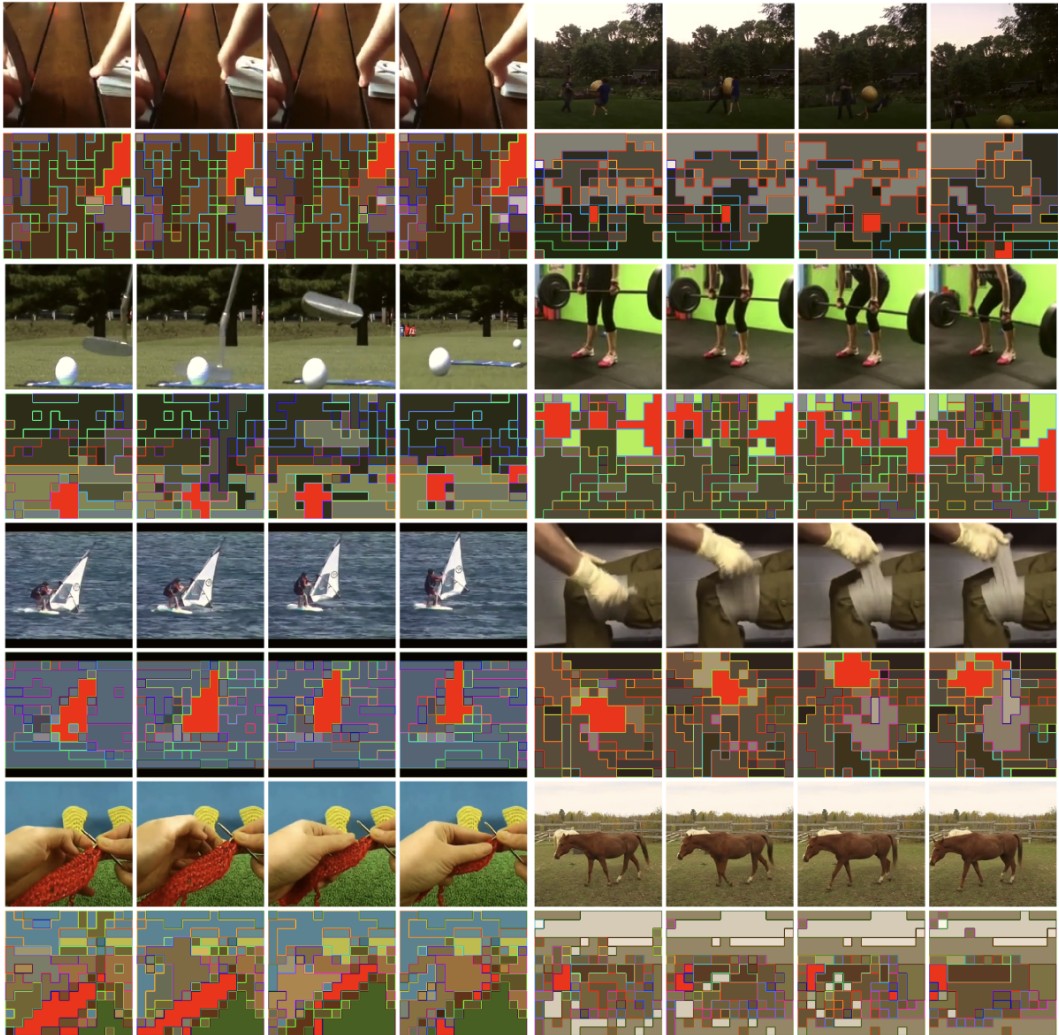

Figure 9: **More visualization on video.** Continuation of Fig. 6. In each clip, we highlight an instance or part being merged into one token across frames (red). Clips are from Kinetics-400 val.

