# OpenReview forum: "Token Merging: Your ViT But Faster"
_ICLR.cc/2023/Conference — ICLR 2023 notable top 5%_

### Official Review · Reviewer_Gshq · 2022-10-14

**Confidence:** 4
**Correctness:** 3
**Technical Novelty And Significance:** 3
**Empirical Novelty And Significance:** 3
**Recommendation:** 6

**Clarity, Quality, Novelty And Reproducibility:**

The novelty of this paper is clear. The paper is well-written as easy to follow.

**Strength And Weaknesses:**

Strength
- The motivation of this paper is clear. For a large ViT model, there is no need to keep all the tokens involved in the decision making as some of them may contain useless information. Discarding some of the tokens that are similar to others can efficiently reduce the inference time.

- The proposed approach is simple and easy to follow.  It can be added in either the training or inference stage. For users who want to use large-scale models but are with limited computational resources, this paper provides an appropriate tool to conquer this issue.

- Experiments show that the performance does not drop much when r is small but the speedup is clear. Moreover, compared to other ViT based models, this paper also shows its advantages.

- Experiments on all images, video, and audio show that the proposed approach indeed behaves well.

- I really like the visualizations in Figure 4. It seems that for regions with similar textures or patterns, they can be merged for efficient inference.

Weaknesses
- Like the author said, the vanilla ViT shows great potential in visual recognition especially after the emerging of MIM-based methods, like MAE and the proposed approach works well with the vanilla ViT. However, in some cases, a pyramid structure of ViT is required especially for downstream tasks, like semantic segmentation. It would be better if the approach could be utilized in ViTs, like Swin Transformer.

- From the experimental results, we can see that for large models, increasing the value of r does not lead to too much performance drop. However, for small-sized or even tiny-sized models, the performance drops much.  Though the reason is clear but I still think it should be mentioned explicitly.

- From Figure 3(a), it seems that after compressing the ViT-L model, the trade-off between the performance and latency is not as good as ViT-B without compression.

**Summary Of The Paper:**

This paper presents a token merging strategy for efficient inference of ViTs. A Bipartite Soft Matching method is proposed. It divides all the tokens into two parts, and draws an edge from each token in the first part to its most similar token in the second part according to the cosine similarity between pairs of tokens. The model efficiency can then be decided by how many similar tokens should be merged.

Different from most previous works that aim at compressing ViTs with training, the approach presented in this paper can be conducted in the inference stage directly. Thorough experiments are performed on images, video, and audio, which reflect the effectiveness of the proposed approach.

**Summary Of The Review:**

The novelty of this paper is clear but not that significant to match the score of 8. The experiments are thorough. I think this paper deserves a score of 6.

---

> ### Author Response · Authors · 2022-11-16
> **Author Response to Reviewer Gshq**
>
> We’d like to thank the reviewer for their thoughtful review. We’ve incorporated their feedback in the updated draft.
>
> **In some cases, a pyramid structure of ViT is required especially for downstream tasks.** We definitely agree! We focus on classification in this paper for simplicity, but this is actually one of the areas where ToMe can work really well compared to something like pruning. Since ToMe can track which tokens got merged, we can actually keep the pyramid structure intact by “unmerging” tokens.
>
> To demonstrate this, we’ve added an experiment to Appendix F of the revised draft where we apply ToMe *without training* to Stable Diffusion [1] for image generation. To perform diffusion, Stable Diffusion uses a UNet model with transformer blocks instead of convs, making it the perfect test for ToMe on a pyramidal architecture (having both downsampling and upsampling pyramids). And by using a simple merge → unmerge strategy described in Appendix F, ToMe can speed up image generation by up to 2x and reduce memory consumption by 3.86x while still producing high quality images. Definitely take a look if you’re interested!
>
>
> **ToMe’s accuracy drop on smaller models.** Fair point, thanks for bringing it up. The accuracy drop without training on small models is definitely higher than for larger models. But this doesn’t mean that ToMe doesn’t work for smaller models. We include trained DeiT-S results in the main paper and trained DeiT-Ti results in the appendix:
>
> | Model   |  r |  acc  |  drop | speedup |
> |---------|:--:|:-----:|:-----:|:-------:|
> | DeiT-S  |  0 | 79.96 |   -   |    -    |
> |         | 16 | 79.13 | -0.83 |  2.00x  |
> | DeiT-Ti |  0 | 71.84 |   -   |    -    |
> |         | 15 | 71.02 | -0.82 |  1.90x  |
>
> Notably, the accuracy drop for both are similar after training: a 0.8% accuracy drop for a 1.9-2x faster model. Interestingly this is nearly on par with our results for ViT-L with ToMe applied during MAE fine-tuning (0.6% drop for a 2x faster model). Thus, we believe ToMe can work well for smaller models given the right training recipe.
>
> We agree that this deserves more discussion in the paper, so we’ve added a sentence to the “Comparing to Token Pruning” paragraph in Sec. in 4.3: “Interestingly, after 300 epochs the DeiT models have a similar accuracy drop to our MAE trained ViT-L (Appendix A).”
>
>
> **From Figure 3(a), it seems that after compressing the ViT-L model, the trade-off between the performance and latency is not as good as ViT-B without compression.** Yeah, this is true for the case where we apply ToMe without training. We see the same result for the MAE models in Figure 3(c) without training. However, once we start training with ToMe, the accuracy becomes better than the ViT-B model (with r=8 being the maximum token reduction for ViT-L):
>
> | Model                         |  acc | im/s |
> |-------------------------------|:----:|:----:|
> | ViT-B                         | 83.6 | 309  |
> | ViT-L + ToMe (r=8 decreasing) | 84.2 | 241  |
> | ViT-L + ToMe (r=8 constant)   | 85.1 | 183  |
> | ViT-L                         | 85.7 | 93   |
>
> Thanks for mentioning this. We realize now that in the original draft this comparison can only be made concretely by cross-referencing with the appendix, which is not ideal. To fix this, we’ve added the ViT-B number to Table 3 in the rebuttal revision.
>
>
> [1] High-Resolution Image Synthesis with Latent Diffusion Models. CVPR 2022.

---

### Official Review · Reviewer_vh1A · 2022-10-24

**Confidence:** 4
**Correctness:** 4
**Technical Novelty And Significance:** 4
**Empirical Novelty And Significance:** 3
**Recommendation:** 10

**Clarity, Quality, Novelty And Reproducibility:**

Clarity: very clear and easy to understand.

Quality: the paper is in high quality and the experimental results are good.

Novelty: the proposed method is novel as far as I can assess.

Reproducibility: the code sample provided in the appendix makes it easy to reproduce.

**Strength And Weaknesses:**

Strengths

+ The proposed method of greedily-merging tokens is very elegant and makes a lot of sense to me, especially that the method can work without retraining. This makes the proposed method easy to use for many off-the-shelf transformer-based models.

+ The paper shows results on three different modalities: images, videos, and audio, and all displayed competitive results. This shows the proposed method is general and robust.

+ The ablation studies in Table 1 are comprehensive, and show the proposed method is robust under different hyer-parameters.

+ The qualitative examples in Figure 4 are interesting. Please make sure they are not cherry-picked.

+ The paper is well motivated, well-written, and easy to understand.

Weaknesses

- The comparison to existing efficient transformers in Table 4 are not as exciting as expected, especially given that DynamicViT is also very simple. Most of the advances are in training speed instead of on speed-accuracy trade-off.

- It will be good to include other efficient transformer methods for videos and audio to provide more context (if there is any) about the numbers.

- [Minor] last line of the code in Appendix D. Should it be `return merge(k)`.?

**Summary Of The Paper:**

This paper works on improving the efficiency of a ViT model by merging tokens each layer. Specifically, the authors first uniformly split ViT tokens into two sets and merge (i.e., replace two tokens with their mean feature) the top k similar pairs across sets in each layer. The merge runs in negligible time and can work on test time without retraining. Experiments on image, video, and audio data show the proposed method can reduce ~50% tokens within <0.5% accuracy drop.

**Summary Of The Review:**

This paper has a simple and effective idea on an important problem, and has great application values. All the technical contributions are well motivated and well ablated. The state-of-the-art comparisons to existing efficient transformer methods (Table 4) are currently a little bit below expectation, but this does not undermine the significance of this work, especially given that the model can work without retraining. My current recommendation is accept, and am happy to raise to strong accept if the authors address my concerns during the discussion session.

---

> ### Author Response · Authors · 2022-11-16
> **Author Response to Reviewer vh1A**
>
> We thank the reviewer for their kind review and useful feedback! We’ve incorporated their feedback in the revised draft:
>
> **“The comparison to existing efficient transformers in Table 4 are not as exciting as expected… Most of the advances are in training speed instead of on speed-accuracy trade-off”.** This is a fair point. However we would like to mention that merging should on paper be much slower than pruning, yet our method is both *faster* and *more accurate* than pruning works. Moreover, as you mention, the main benefit of ToMe is its *versatility*. Not only can it be used without retraining, but it can also be used in situations where pruning cannot (e.g., dense prediction tasks like segmentation, detection, and image generation).
>
> To emphasize both of these points, we’ve added an experiment to Appendix F of the revised draft with ToMe being applied *without training* on Stable Diffusion [1] for image generation. For this task, training your own model would be prohibitively expensive, and because we’re predicting noise for every pixel in the image, we can’t afford to remove information with pruning.
>
> In this experiment, we show that with a very simple application of ToMe (4 lines of code) we can speed up image generation by up to 2x and reduce memory consumption by 3.86x while still producing high quality images. Conversely, we find that pruning even just 10% of the tokens in the image significantly reduces image quality, making this a task where ToMe significantly outperforms pruning. Please take a look if you are interested!
>
>
> **More context for efficient transformers in video and audio.** Good point! To our knowledge, we are the first token reduction work to evaluate on audio. However, there is one other recent work that evaluates on video: Adaptive Token Sampling [2] (ATS). Here is the relevant comparison on k400:
>
> | Model              |  acc |  gflops  | speed-up |
> |--------------------|:----:|:--------:|:--------:|
> | X-ViT (16x)        | 80.2 | 425x1x3  |     -    |
> | X-ViT (16x) + ATS  | 80.0 | 259x1x3  |   1.64x  |
> | ViT-L              | 84.7 | 598x1x10 |     -    |
> | ViT-L + ToMe (r65) | 84.5 | 281x1x10 |   2.13x  |
>
> They use X-ViT instead of ViT-L, but ATS has a result on k400 where they only have a 0.2% accuracy drop so we can do a comparison here. In terms of flop reduction, ToMe gets the same accuracy drop with a 2.13x speed-up as ATS does with a 1.64x speed-up. But more importantly, ATS requires training to get this result while this ToMe result is without training.
>
> Thanks for bringing this up. We’ve included the X-ViT (16x) + ATS model to Table 5.
>
>
> **Should the last line of Appendix D be return merge(k)?** Thanks for reviewing the code! We tried to make the implementation re-usable, so the function returns a lambda that you would then apply to any input you want (including x). This allows us to easily implement token source and size tracking, since we can just use the same merge lambda output by this function. It’s a bit odd to return a function here, but it works really cleanly in practice! We’ll be releasing the full code for all settings later.
>
>
> [1] High-Resolution Image Synthesis with Latent Diffusion Models. CVPR 2022.
> [2] Adaptive Token Sampling For Efficient Vision Transformers. ECCV 2022.

---

> > ### Comment · Reviewer_vh1A · 2022-11-30
> > **Thank you for your response**
> >
> > Thank the authors for providing the rebuttal. My only concern on comparison to other efficient video transformer is resolved by the additional numbers in the rebuttal. I do not see more concerns from other reviewers. I am glad to raise my rating to strong accept and hope this paper can be highlighted in the conference.

---

### Official Review · Reviewer_eArF · 2022-10-25

**Confidence:** 4
**Correctness:** 3
**Technical Novelty And Significance:** 3
**Empirical Novelty And Significance:** 3
**Recommendation:** 8

**Clarity, Quality, Novelty And Reproducibility:**

 - This paper has a nice writing quality and presentations, except for some slightly-incorrect statement:

e.g. At Page 1 Introduction: "despite being overtaken in cost to performance, vanilla ViTs still have many desirable qualities". from my perspective, the statement here "being overtaken in cost to performance" might not be correct. As ViTs have many nice properties (also mentioned by the authors) and work well at many tasks, I would like the authors to rephrase this sentence.

**Strength And Weaknesses:**

#### **Strength**

 - The problem, reducing the computational burden in ViTs, has been an important issue in developing and deploying transformers.

 - While redundancy in self-attention has already been investigated before, there are few solutions except for pruning-based methods, which require additional attention. The simple strategy in this work shows promising results, even without being trained.

 - The authors perform concrete studies and experiments across different tasks and architectures, showing the reliability ToMe.

#### **Weakness**

 - This method could be viewed as a parallel implementation of the matching algorithm. We expect more discussion with previous slower variants, such as the clustering algorithm in [1].

 - The authors mentioned that the method could be used in training to reduce training complexity. One more common use case here is to adopt such idea to recent MIM-based methods such as MAE, where longer pre-training epochs are required.  However, the authors have only used it in fine-tuning MAE pre-trained checkpoints.

[1] Token Pooling in Vision Transformers. arxiv 2110.03860.

**Summary Of The Paper:**

 - This work proposes to reduce the computational redundancy in vision transformers (ViT) with Token Merging, without significantly changing the original architecture.

 - With the introduced ToMe module, similar tokens are gradually combined with a simplified matching algorithm. The author also perform studies on the comparison with pruning.

 - Extensive experiments on image and video recognition are conducted, across a variety of transformer families.

**Summary Of The Review:**

 - Reducing the computational burdens in vision transformers is a non-trivial task. Given the simplicity and effectiveness of ToMe, I believe it's beneficial in further research.

---

> ### Author Response · Authors · 2022-11-16
> **Author Response to Reviewer eArF**
>
> We thank the reviewer for their review and appreciate their feedback! We’ve incorporated this feedback as described below:
>
> **Comparison with Token Pooling [1].** Yes, this is definitely a very important comparison. All of the methods in [1] derive from k-means, and in our setting k-means performs poorly on top of being slow (results for off-the-shelf ViT-L with maximal token merging):
>
> | algorithm            |  acc  |  im/s |
> |----------------------|:-----:|:-----:|
> | Baseline             | 85.96 | 93.3  |
> | ToMe (ours)          | 84.25 | 182.9 |
> | k-means (2 iter) [1] | 80.19 | 169.7 |
> | k-means (5 iter) [1] | 80.29 | 147.5 |
>
> The authors of [1] don’t release code so we couldn’t compare to the more complicated algorithms in [1] using our setting. However, we can always compare in their setting: in their appendix they report an over 10% accuracy drop with their best method (WK-Metroids) on DeiT-S for half the flop count without training (Figure 9 in [1] @ 2.3 gflops). In a similar setting our accuracy drop is only 4.4% (Table 8b, ViT-S/16 r=16).
>
> We agree that the main paper was lacking some of these points. We’ve revised Sec. 4.1 “Comparing Matching Algorithms.” to take into account this discussion.
>
> **ToMe could be used during MAE Pretraining.** Great point! We were thinking the same, and we’re actively exploring this area. As a sneak peek, we have preliminary experiments where using ToMe during MAE pretraining reduces the compute cost while keeping the accuracy exactly the same. The trick is to select the correct merging rate for the given masking ratio. Keep an eye out for future work in this area!
>
> **Page 1 Introduction: the statement here "being overtaken in cost to performance" might not be correct.** Thanks for pointing this out! After giving it more thought, we agree that this statement was incorrect. ViTs are indeed consistently getting better as more effective training techniques are introduced (most recently DeiT III [2]), managing to stay competitive with newer models despite the architecture staying largely the same. We’ve removed "being overtaken in cost to performance" from the introduction.
>
> [1] Token Pooling in Vision Transformers. arxiv 2110.03860.
> [2] Deit III: Revenge of the ViT. ECCV 2022.

---

### Official Review · Reviewer_MRoC · 2022-10-29

**Confidence:** 4
**Correctness:** 4
**Technical Novelty And Significance:** 3
**Empirical Novelty And Significance:** 3
**Recommendation:** 8

**Clarity, Quality, Novelty And Reproducibility:**

Very clear and the quality is up to the bar of ICLR.
Token pruning methods are considered a lot in the literature, this paper drafts the story centering on token merging instead, which is novel to me.

**Strength And Weaknesses:**

Strength:

* The idea is simple yet effective and can benefit both ViT training and inference a lot.
* The authors clearly have deep insights about the (larger or smaller) ViT designs. The introduction of the literature is clear and informative.
* Using matching instead of clustering algorithm makes sense to me, results also demonstrate the expected throughput achievements.

Weakness:
* More direct comparisons are desired, e.g., in Tab. 3, you compare ViT-L with other ViTs. How about comparing them apple-to-apple. E.g., ViT-L w/ ToMe vs. ViT-L (attached in appendix; better to remove ahead), MViTv2 w/ ToMe vs. MViTv2, Swin w/ ToMe vs. Swin, etc.
* It may be easier to improve the throughput for large ViT models without hurting the model accuracy a lot. How about smaller ones, e.g., LeViT?

**Summary Of The Paper:**

This paper proposes a simple method to merge tokens within ViTs with or without training, increasing the throughput by up to 2x across various image, video, and audio tasks. ToMe inherits the spirit of simplicity and can server as a drop-in replacement to increase the training speed or inference speed.

**Summary Of The Review:**

In short, the idea is interesting and work for ViTs. A lot of visualizations also intuitively explain and validate the idea. Thus I recommend to accept this work.

As for its generalizability, more experiments on both small models and large models can be considered as the cost is not so high.

---

> ### Author Response · Authors · 2022-11-16
> **Author Response to Reviewer MRoC**
>
> We thank the reviewer for their positive comments and helpful feedback! We’ve incorporated the reviewer’s feedback as described below:
>
> **ViT-L w/ ToMe vs. ViT-L.** We agree that this is an important comparison. Thus we include this comparison as well as ViT-H w/ ToMe vs. ViT-H in Table 3. We mark the ViTs without ToMe in gray to denote that they are the baseline to compare with the ToMe models. However, this wasn’t properly made clear in the table caption, so thanks for mentioning this. We’ve updated the caption in the revised draft to make this clear.
>
> **Does ToMe work as well on tiny models?** This is a very good question! For smaller models, we include DeiT-S in the main paper and DeiT-Ti in the appendix (Table 11), which has fewer parameters than the smallest LeViT model:
>
> | Model   |  r |  acc  |  drop | speedup |
> |---------|:--:|:-----:|:-----:|:-------:|
> | DeiT-S  |  0 | 79.96 |   -   |    -    |
> |         | 16 | 79.13 | -0.83 |  2.00x  |
> | DeiT-Ti |  0 | 71.84 |   -   |    -    |
> |         | 15 | 71.02 | -0.82 |  1.90x  |
>
> Notably, the accuracy drop for both are similar after training: a 0.8% accuracy drop for a 1.9-2x faster model. Interestingly this is nearly on par with our results for ViT-L with ToMe applied during MAE fine-tuning (0.6% drop for a 2x faster model). Thus, we believe ToMe can work well for smaller models given the right training recipe. Thanks for pointing this out, we forgot to mention the DeiT-Ti experiment in the main paper so we’ve added a reference in Table 4.
>
> **ToMe with other Transformer Architectures.** Thanks for this suggestion! Applying ToMe to other Transformer architectures is an exciting direction that we have been exploring for future work. Note that architectures like Swin, MViT, and LeViT have specialized components that need to be considered when implementing token merging (e.g. shifted window in Swin, relative position embedding in MViTv2, and attention bias in LeViT), so we focus on ViT in this paper.
>
> However, to show that ToMe can work with other architectures and tasks, we include a new experiment in Appendix F of the revised draft with ToMe applied to Stable Diffusion [1] for image generation without training. We find that even with this drastically different architecture and task, ToMe can speed up image generation by up to 2x and reduce memory consumption by 3.86x while still producing high quality images. Please take a look if you are interested.
>
> [1] High-Resolution Image Synthesis with Latent Diffusion Models. CVPR 2022.

---

> > ### Comment · Reviewer_MRoC · 2022-11-24
> > **Response**
> >
> > Thank the authors for the feedback. After reading your rebuttal, I would like to maintain my score and recommend acceptance.

---

### Public Comment · ~Zizheng_Pan1 · 2023-02-14
**Great work!**

Dear authors,

Congratulations on this excellent work! It is very impressive to see the proposed Token Merging can speed up off-the-shelf ViTs by ~2x on ImageNet classification, even without training! Besides, I would like to mention that the proposed Token Merging is related to Deformable Token Merging (DTM) in LIT [A], where we dynamically merge tokens based on objects’ scales and shapes. It would be beneficial to include a discussion to it. Thank you!

[A] Pan, Zizheng, et al. "Less is more: Pay less attention in vision transformers." Proceedings of the AAAI Conference on Artificial Intelligence. Vol. 36. No. 2. 2022.

---

> ### Author Response · Authors · 2023-02-26
> **Thanks for your comment!**
>
> Thanks for bringing this up! We've added a sentence discussing the deformable token merging module in LIT to our camera ready's related work.

---

### Decision · Program_Chairs · 2023-01-20

**Decision:**

Accept: notable-top-5%

**Justification For Why Not Higher Score:**

N/A

**Justification For Why Not Lower Score:**

The proposed method is very simple, yet very effective. The AC believes this work will be of significant interest to a wide ICLR audience and practitioners in the field.

**Metareview: Summary, Strengths And Weaknesses:**

The paper proposes a novel method for increasing the throughput of transformers (without training), by progressively merging similar tokens in the model, instead of pruning them. The proposed method is very simple yet very effective, yielding up to 2x speed up across image, video, and audio classification. The motivation is clear, the experimental analysis is comprehensive, and the obtained results are compelling. A few weakness have been pointed out by the reviewers, including the lack of application of the method in modern/specialized transformer architectures such as Swin, MViT, etc. and during MAE pretraining. Overall, this is a good paper. All reviewers recommend acceptance and the AC agrees with this decision.

**Note From Pc:**

if the above contains the word "oral" or "spotlight" please see: "oral" presentation means -> notable-top-5% and "spotlight" means -> notable-top-25%. As stated in our emails, we are disassociating presentation type from AC recommendations

**Summary Of Ac-Reviewer Meeting:**

N/A